# IgA Vasculitis and IgA Nephropathy: Same Disease?

**DOI:** 10.3390/jcm10112310

**Published:** 2021-05-25

**Authors:** Evangeline Pillebout

**Affiliations:** 1Nephrology Unit, Saint-Louis Hospital, 75010 Paris, France; evangeline.pillebout@aphp.fr; 2INSERM 1149, Center of Research on Inflammation, 75870 Paris, France

**Keywords:** IgA Vasculitis, IgA Nephropathy, adults, children, presentation, physiopathology, genetics, prognosis, treatment

## Abstract

Many authors suggested that IgA Vasculitis (IgAV) and IgA Nephropathy (IgAN) would be two clinical manifestations of the same disease; in particular, that IgAV would be the systemic form of the IgAN. A limited number of studies have included sufficient children or adults with IgAN or IgAV (with or without nephropathy) and followed long enough to conclude on differences or similarities in terms of clinical, biological or histological presentation, physiopathology, genetics or prognosis. All therapeutic trials available on IgAN excluded patients with vasculitis. IgAV and IgAN could represent different extremities of a continuous spectrum of the same disease. Due to skin rash, patients with IgAV are diagnosed precociously. Conversely, because of the absence of any clinical signs, a renal biopsy is practiced for patients with an IgAN to confirm nephropathy at any time of the evolution of the disease, which could explain the frequent chronic lesions at diagnosis. Nevertheless, the question that remains unsolved is why do patients with IgAN not have skin lesions and some patients with IgAV not have nephropathy? Larger clinical studies are needed, including both diseases, with a common histological classification, and stratified on age and genetic background to assess renal prognosis and therapeutic strategies.

## 1. Introduction

Since the first descriptions, in the last century, of IgA Nephropathy, at the time called Berger’s disease, and IgA vasculitis called Henoch-Schönlein Purpura, the authors suggested that they were two clinical manifestations of the same disease. Recent advances in understanding pathophysiological mechanisms of these two entities have only reinforced this idea. Nevertheless, scientific arguments confirming this hypothesis are scarce.

Many, rather old, clinical cases have been published. They describe episodes of one or the other disease within the same siblings, in particular in homozygous twins [1], simultaneously or successively [2], in a father and son [3], in the same patient at two periods of his life [4,5,6,7] and, more recently, recurrences in kidney allograft in one form or another [8,9,10].

Four rather exhaustive reviews have been published [1,11,12,13], collecting these cases and gathering data over time to argue in favor of the hypothesis of a common disease. The last review dates from ten years ago. Since then, some clinical studies have been issued, but they are unfortunately too rare.

Indeed, only clinical studies involving patients with IgA vasculitis (IgAV) or IgA Nephropathy (IgAN) within the same cohort would reveal differences or similarities in terms of epidemiology, presentation, prognosis, sensitivity to treatment, physiopathology, biomarkers, and genetics.

## 2. Epidemiology

Apart from the obvious differences in extra-renal clinical presentation, the main difference between IgAV and IgAN is epidemiological (Table 1). The overall precise prevalence of IgAN or IgAV, in children or adults, is not known, but varies considerably around the world [14].

In children, IgAV incidence is about 3 to 26.7/100,000 children per year [15,16,17]. The risk of progression to end-stage renal disease requiring dialysis in children varies from 2.5% to 25%, but it is on average around 8% [18,19]. In Europe, 3% of children on dialysis are due to IgAV. It is much rarer in adults, where its incidence is around 1.4 to 5.1/100,000 [20,21,22,23]. The child/adult ratio would therefore vary from 150 to 205. In adults, the risk of developing chronic renal failure is frequent, from 8 to 68%, on average around 18% within 15 years [24,25].

In children, IgAN incidence varies from 0.03/100,000/year in Venezuela to 9.9/100,000/year in Japan and in adults around 2.5/100,000/year [26,27]. The reported incidence rates are likely to underestimate true rates of IgAN, as this disease can exist sub-clinically and depends on country policy for microscopic hematuria detection in the population and/or renal biopsy indications. The risk of progression to end-stage renal disease (ESRD) is reported from 4% after 4.6 years in Europe [28] to 11% after 15 years of follow-up in Japan [29] and 14% in China [30]. The risk of progression to ESRD requiring dialysis in adults varies from 30% to 40% within 10 to 25 years [31]. In adults, IgAN is the cause of ESRD of 3.6% of newly dialyzed patients [32].

Whatever the studies, there is definitely an age difference between patients with IgAN and those with IgAV. This is particularly demonstrated by this Japanese study [33]: Among 18,967 patients with biopsy-proven disease between July 2007 and December 2012, the authors selected 513 patients with IgAV and 5679 with IgAN from the J-RBR (Japan Renal Biopsy registry) registry. They highlight a bimodal distribution for IgAV with two peaks between 1 to 19 years and 60 to 69 years, whereas for IgAN, the peak is rather between 30 to 39 years.

## 3. Clinical Presentation and Outcome

Macroscopic hematuria is the most frequent IgAN initial presentation in children, followed by the fortuitous finding of microscopic hematuria accompanied or not by proteinuria. In adults, the diagnosis is often made at the stage of chronic kidney disease (CKD), as if diagnosis in this belated context were made at an advanced stage after missing a pauci-symptomatic form that previously occurred in childhood. By contrast, the diagnosis of IgAV is, in most cases, made much earlier, revealed by the presence of extra-renal signs.

Indeed, by definition, the IgAV is characterized by the combination of cutaneous (palpable purpura), gastrointestinal (colicky pain, bloody stools) and articular (arthralgia) involvement. More rarely, we can observe a neurological, pulmonary or urological involvement [34]. The long-term prognosis of the disease depends on the presence of renal impairment and its evolution. From a histological point of view, it is not possible to distinguish a glomerulonephritis as part of an IgAV from an IgAN. Renal biopsy shows, in the two cases: on immunofluorescence, predominant IgA1 deposits in the mesangium of all glomeruli (Figure 1), with glomerular deposits of IgG, IgM, C3 and fibrin in variable proportions; on light microscopy, mesangial hypercellularity with increased mesangial matrix, endo-capillary hypercellularity, segmental glomerular sclerosis, cellular crescents and tubular atrophy and interstitial fibrosis.

IgAV is most common in childhood, but may occur at any age. Macroscopic hematuria is very uncommon after age 40. The importance of asymptomatic urine abnormality as the presentation of IgAN will depend on attitudes to routine urine testing and renal biopsy. It is unclear whether patients referring late, with chronic renal impairment, have a disease distinct from those referring earlier, with macroscopic hematuria (Figure 2: Data from patients presentation in Leicester, UK, 1980–1995 [35]).

Renal involvement occurs in IgAV with a prevalence ranging from 20 to 54% in children and 45 to 85% in adults. Patients with nephritis have hematuria ± proteinuria of varying flow rate. They seem to have a higher frequency of nephritis and rapidly progressive nephritic syndrome at onset, notably in adults, but this is not the case in all published cohorts, as some of them, as we will develop furthermore, even describe the opposite.

Whether or not disease severity at diagnosis and prognosis differs between IgAN and IgAV remains controversial.

With regard to the clinical presentation and prognosis, the results of the few studies comparing the two diseases most frequently show a less favorable evolution for IgAN. Significant differences in the conclusions of the studies from one country to another suggest genetic susceptibility.

A Chinese study [36] compared 31 children with IgA Nephropathy to 120 children with IgA vasculitis and nephropathy, 32 of whom had a renal biopsy. In this pediatric cohort, patients with IgAN were significantly older. Histologically, the kidneys of patients with IgAN were more sclerotic (35.5% versus 3.1%) while there were more endothelial proliferation in those with IgAV (65.6% versus 29%). After 34 months of follow-up, the prognosis seemed better in patients with IgAV, since 72.5% of children from the IgAV group were in complete remission compared with only 19.4% in the IgAN group, in which patients had significantly persistent hematuria and proteinuria.

In another Spanish study [37], 142 patients with IgAV were compared to 61 patients with IgAN, of all ages. Those with IgAV were also younger (mean age at onset 30.6 years (2.9 to 82.7 years), than those with IgAN (37.1 years (14.7 to 78.5 years)). Again, the renal presentation was less severe: a lower rate of renal insufficiency (25% in IgAV vs. 63.4% in IgAN) and nephrotic syndrome (12.5% vs. 43.7%) were reported, while the prognosis, after a median follow-up of 130 months, was better: dialysis (2.9% in IgAV vs. 43.5% in IgAN), renal transplant (0% vs. 36%) and residual chronic renal insufficiency (4.9% vs. 63.8%)

In a Korean study [38], 92 adult patients with kidney biopsy-proven IgAV were compared to 1011 adult patients with kidney biopsy-proven IgAN, 89 and 178, respectively, from each group were matched using a propensity score in order to compare long term renal outcome. Once again patients with nephropathy in the context of IgAV were younger (33.2 ± 15.9 years vs. 37.7 ± 13 years), and less frequently developed renal insufficiency (7.6% vs. 14.4%) and interstitial fibrotic lesions on biopsy: however, these differences disappeared once the propensity score was applied to the subgroup. In addition, while the renal prognosis seemed better in the IgAV group (ESRD 14.1% vs. 22%; remission 28.3% vs. 13.8%), this difference disappeared (ESRD 14.6 vs. 13.5%; remission 28.1% vs. 27.5%) in the cohort of patients matched for clinical, histological and treatment factors. In my opinion, this study is essential, and it is the one that more clearly sheds light on the matter. It substantiates that the worse prognosis ascribed, from the other above-mentioned studies, to IgA Nephropathy, is probably pointed out only because we selected the most severe patients, namely, those who had not recovered spontaneously.

A Chinese pediatric study [39] compared the clinical-histological presentation of 41 patients with IgAN and 137 with IgAV. Patients with IgAV had higher levels of blood white cell, hemoglobin and platelet, and lower levels of hematuria, blood nitrogen and C4, compared to IgAN cases, but without any clinical and histological renal significant difference. The authors did not perform any prospective follow-up.

Another Japanese study [33] compared 513 patients with IgAV and 5679 with IgAN, of all ages, from a renal national biopsy registry. As previously mentioned, the age at diagnosis considerably differed: IgAV peaked at 1 to 19 years and 60 to 69 years, whereas IgAN peaked at 30 to 39 years. In contrast to previous studies, it appears that, regardless of age, patients with IgAV had a more severe clinical presentation with more frequent rapidly progressive renal insufficiency (4.5% vs. 1.4%) and a nephrotic syndrome (10.5% vs. 3%), and biopsies showed more inflammatory lesions including more endocapillary proliferation (6.4% vs. 0.9%) and extra-capillary (6.6% vs. 0.8%) glomerulonephritis. In contrast, patients with IgAN, had significantly more chronic nephritic syndrome (88.5% vs. 61.6%). Unfortunately, this cross-sectional study had no reference to the renal outcomes for any of the patients.

A further Japanese study [40] analyzed cross-sectionally 24 IgAV adult patients with nephritis or 56 adult patients with IgAN, all of whom underwent renal biopsy. The clinical characteristics did not differ between groups. Duration from onset was significantly longer for IgAN (47.6 months vs. 8.8 months). More patients with IgAV received steroid therapy, whereas more patients with IgAN received renin-angiotensin-aldosterone system (RAAS) blockers. Compared to IgAN, the mean rate of global sclerosis or crescent formation were significantly lower in patients with IgAV. Using Oxford classification, IgAV patients had more endothelial injury and IgAN worse mesangial proliferation, crescent formation, and tubulointerstitial injury.

A French retrospective study [41] showed that the risk score of end-stage renal disease or death, including hypertension, proteinuria more than 1 gramme per day and severe pathological lesions (local classification score ≥ 8), validated in a cohort of 1064 patients with an IgAN, was also valid when it was applied to a subgroup of 74 patients with IgAV from this same cohort, after 8.2 years of follow-up.

More recently, the National Institute of Diabetes and Digestive and Kidney Disease (NIDDK) funded the CureGN study to establish a primary glomerular disease consortium with a focus on IgAN and IgAV. Using this longitudinal observational cohort of adults and children with biopsy-proven primary glomerular disease, from all over the world, this first report described the baseline clinical characteristics of the two nosological entities [42]. The next step will be to better predict the long-term follow-up renal outcome, find prognostic biomarkers and identify patients most appropriate for specific therapies. A total of 667 patients were enrolled, including 506 (75.9%) with IgAN and 161 (24.1%) with IgAV, 285 (42.7%) were children. At the moment of biopsy, patients with IgAV were younger, more frequently white, and had a higher estimated glomerular filtration rate and lower serum albumin than those with IgAN. Adults and children with IgAV were similar in terms of proteinuria, hematuria or serum albumin. Patients with IgAV were more likely to be treated with immunosuppressive therapy if compared to those with IgAN, but less likely to receive standard supportive care with RAAS inhibition. No data is available to date about renal histology.

## 4. Treatment

The Kidney Disease Improving Global Outcome (KDIGO) working group on glomerulonephritis compiled the first evidence-based guidelines for the treatment of IgAN and IgAV in 2012 [43]. Children and adults were treated in the same way.

In adults and children with IgAN, the particular value of RAAS blockers, as angiotensin-converting-enzyme inhibitors [44,45] or angiotensin-receptor blockers [46] in retarding progression of the disease, has been shown in prospective randomized trials. Supportive care is now the first line of treatment, recommended in IgAN and IgAV, as soon as proteinuria is > 1 g per day (in children 0.5 to 1 g/day per 1.73 m^2^) with a blood pressure goal < 130/80 mmHg (no BP goals specified for children), and to achieve proteinuria < 1 g per day.

Concerning the use of corticosteroids and other immunosuppressive agents, KDIGO, in 2012 [43], published its guidelines for IgAN and IgAV, in children as in adults, with a quiet low level of proof (no more than 2C) because of the lack of large clinical trials:

The KDIGO practice guidelines have been recently update [47], including several important clinical trials for IgAN published since 2012, either with corticosteroids or other immunosuppressive drugs, mainly mycophenolate mofetil (MMF). Unfortunately, all have excluded patients with IgAV (adult or children) and children with IgAN. No additional recommendation was done for them.

The authors stressed that future guideline recommendations will need to include an assessment of the relative risks and benefits of steroids in individual patients over a broader range of eGFR, with careful consideration of infections and prophylaxis. They emphasize that we must pay particular attention to comorbidities as advanced age, metabolic syndrome, morbid obesity, latent infection as viral hepatitis or HIV, active peptic ulceration or uncontrolled psychiatric illness.

Concerning corticosteroids in adult IgAN, three main clinical studies [48,49,50] and one meta-analysis [51] are available. They showed that corticosteroids, after optimization supportive treatment (mainly RAAS blockers) and in addition to it, can decrease proteinuria and slow loss of kidney function, particularly in patients with persistent proteinuria more than 1 g/g and preserved renal function (eGFR > 50 mL/min/1.73 m^2^). The Supportive Versus Immunosuppressive Therapy for the Treatment of Progressive IgA Nephropathy (STOP-IgAN) study recently published its follow-up data, available for 149 participants, with a median of 7.4 years, which showed no difference of renal outcomes (in terms of serum creatinine, proteinuria, end-stage kidney disease, and death) between the two groups [49,52]. The Therapeutic Evaluation of Steroids in IgA Nephropathy Global Study (TESTING) trial was stopped early after an interim analysis revealed a high risk of infectious serious adverse events including the lethal Pneumocystis Jirovecii pneumonia [50]. Patients with IgAN from the European Validation Study of the Oxford Classification of IgAN (VALIGA) cohort, classified according to the Oxford-MEST classification and medication used, have been retrospectively studied. From the 1147 patients of the cohort, 184 subjects who received corticosteroids and RAAS blockers were matched to 184 patients with a similar risk profile of progression who received only RAAS blockers. Using a propensity score, authors showed that corticosteroids reduced proteinuria and the rate of renal function decline and increased renal survival, even in patients with an eGFR < 50 mL/min per 1.73 m^2^ [48].

Although previous studies [53,54] showed that MMF was not effective for treatment of IgAN, recent trials [55,56] add conflicting information. Hou JH and al.’s study reintroduces the possibility that MMF may be useful for IgAN, notably by its steroid-sparing effect.

In adult IgAV, very limited evidence is available regarding the value of corticosteroids, cyclophosphamide or other immunosuppressive agents. Only one RCT is available and shows, in adults with severe IgAV, most of them with nephritis, no additional benefit when cyclophosphamide was added to corticosteroids [57,58]. However, by extrapolating from findings in adults with IgAN, the KDIGO guidelines suggest that a 6-month course of corticosteroid therapy would be effective to treat nephropathy in IgAV, if proteinuria is more than 1 g per day persisting despite RAAS blockade and blood pressure control, whereas it has been shown to be ineffective in IgAN [59], Rituximab seems to be a promising therapy in the management of adults with IgAV [60].

In child IgAV, although earlier studies showed some benefit of corticosteroids, these studies were small and poorly designed. Meta-analyses [61] and more recent randomized controlled trials [62,63] have not clearly shown its benefit and confirm that cyclophosphamide therapy is ineffective in severe renal disease. Despite that, based on the opinions of 16 experts in pediatric rheumatology, systemic vasculitis, and nephrology across Europe, the recent SHARE [64] (Single Hub and Access Point for Paediatric Rheumatology in Europe) initiative recommends corticosteroids to be used for treatment of IgAV nephritis in children, regardless of severity. For severe nephritis, the authors drew on experiences with similar forms of systemic vasculitis to recommend intravenous cyclophosphamide in combination with high-dose steroid therapy to induce remission, followed by azathioprine or MMF in combination with low-dose steroid therapy as a maintenance treatment.

Two randomized placebo-controlled prednisone trials [62,65] and one meta-analysis [61] showed that corticosteroids are ineffective to prevent occurrence of nephritis in children with IgAV. One ongoing study (NCT04008316) will evaluate colchicine in adult patients with IgAV limited to skin to prevent skin relapse (primary endpoint) and occurrence of digestive or kidney involvement (secondary endpoint).

A great quantity of clinical trials concerning IgAN are ongoing (about one hundred registered in ClinicalTrial.gouv), with four molecules particularly attractive (sparsentan, hydrochloroquine, budesomide, glifozine). None of them included IgAV or children. It is therefore not possible to compare the sensitivity to treatment of both diseases in those two populations.

## 5. Physiopathology

In recent years, considerable progress has been made in understanding the physiopathology of IgAN. The multi-hits hypothesis is now recognized [66]. The first hit comprises the increased level of circulating galactose-deficient (Gd)-IgA1, influenced by both environmental and genetic factors. Second, antibodies (IgA or IgG) recognizing Gd-IgA1 are produced or already present, possibly attributable to molecular mimicry. The formation of Gd-IgA1-containing immune complexes is thirdly mediated by complement factors and IgA receptors as the soluble IgA Fc alpha receptor (FcαR/sCD89), transglutaminase2 (TG2) and transferrin receptor (TfR/CD71) and fourth, Gd-IgA1 containing immune complexes deposit in the mesangium, hereby inducing a proliferation of mesangial cells and an overproduction of extracellular matrix components, cytokines (Interleukins, tumor necrosis factor-alfa (TNF-α), tumor growth factor-beta-1 (TGFβ-1)) and chemokines (monocyte chemotactic protein-1 (MCP-1)), which ultimately leads to renal dysfunction (Figure 3).

Other pathophysiological pathways are currently being studied, in particular with the aim to develop new treatments. The Toll-like receptors (TLRs) family plays a critical role in the mammalian innate immune system, particularly with regard to the mucosal immunity, which plays a key role in initiating the pathogenic process in IgAN, and it is the first line of host defense against invading pathogens. Activation of TLR-mediated signaling pathways induces gene expression of inflammatory cytokines and type I interferon [67,68]. The complement system, as well, is part of the innate immune system which can enhance the clearance of microorganisms. Skin and mesangial deposits in IgAV and IgAN contain the complement components C3 and C5-C9. Serum levels of activated C3 and mesangial C3 deposition correlate with loss of renal function. The degree of complement activation is also important and have been shown to have prognostic value [69,70,71,72,73].

## 6. Biomarkers

The search for diagnostic and prognostic biomarkers is based on the above-mentioned physio pathological mechanisms. Finding biomarkers able to identify, at an early stage, patients at risk of renal progression, those who will need treatment, is a challenge.

Most of these studies focused on either IgAN or IgAV, but rarely both in the same study, as shown in Table 2.

**Table 2 jcm-10-02310-t002:** Potent value of biomarkers correlated to clinical and/or histological activity and outcome evaluated in IgAN and IgAV or both [ref].

	IgAN	IgAV	IgAN + IgAV
GdIgA1	[74,75]	[76,77,78]	[40,79]
GdIgA1/sCD89	[80]	[77,78]	
GdIgA1/IgG	[81]	[77,78]	
sCD89			[82]
Transglutaminase2			[83]
CD71			[83]
TLR9	[84,85]	[68]	
TLR4			[67,68]
TGF-β1 MCP1			[40]
Complement system	[70,73]	[86]	

Nevertheless, it is now accepted that the aberrantly glycosylated IgA1 (GdIgA1) plays a central role. Several methods are available to highlight and quantify GdIgA1. Mass spectrometry is the reference method, but it is difficult to use in current practice. The Helix aspersa agglutinin (HAA) lectin method is the most commonly used but lacks sensitivity, specificity and reproductivity [87]. Recently, the Japanese Suzuki team has developed a novel lectin-independent method with a specific monoclonal antibody (KM55 mAb) for measuring serum level of GdIgA1, that is clearly more performant and robust [88]. This team first showed that serum IgA, GdIgA1, and immune complexes containing GdIgA1 were increased in patients with IgAN but not in healthy subjects. The IgG anti-GdIgA1 antibody was particularly efficient in this study to make the diagnosis of IgAN since its sensitivity was 89% and its specificity was 92% (compare to the reference test HAA lectin method).

In IgAV, GdIgA1 (measured by HAA lectin method) and immunes complexes containing GdIgA1 are also increased in case of nephritis in children [78] and adults [77]. The team then used their GdIgA1-specific monoclonal antibody KM55 to highlight GdIgA1 in the glomeruli of 48 patients with IgAN and 14 patients with IgAV while it remains undetectable in the glomeruli of 35 with other glomerulopathies in which glomerular IgA deposit is frequent (lupus nephritis, HCV-related nephropathy, mesangio-proliferative glomerulonephritis, membranous nephropathy, hepatic glomerulosclerosis) [74].

Another team analyzed adult patients with renal-biopsy proven IgAN and IgAV. Serum GdIgA1 levels and glomerular GdIgA1 staining, using enzyme-linked immunosorbent essay (ELISA) with the same anti-human GdIgA1 specific monoclonal antibody (KM55), were comparable among patients with IgAN and IgAV [40].

## 7. Genetics

Familial clustering, ethnic differences, and regional discrepancies suggest a genetic component to IgAN and IgAV.

The genetic influence of GdIgA1 production and expression during IgAN and IgAV is highlighted by a study from Kiryluk et al [89]. Serum GdIgA1 levels (quantified using HAA lectin based ELISA) from 20 children with IgAV and nephritis, and 14 children with IgAN were compared to 51 age- and ethnicity match pediatric controls. Serum level of GdIgA1 were significantly elevated in the 34 children with IgAV or IgAN compare to controls. It was also elevated in a large fraction of 54 first-degree relatives, compared with 141 unrelated healthy adult controls. A unilineal transmission of the trait was found in17, bilineal transmission in 1, and sporadic occurrence in 5 of 23 families when both parents and the patient were analyzed. There was a significant age-, gender, and household-adjusted heritability of serum GdIgA1 estimated at 76% in pediatric IgAN and at 64% in pediatric IgAV with nephritis.

High heritability of GdIgA1 in IgAN have been previously shown in adults from other ethnic origins: Caucasiens [90], Asians [91], and African Americans [92].

Kiryluk et al. collaborated with many teams around the world to provide insight into why IgAN differs in terms of incidence, presentation and prognosis all over the world, through a huge genome-wide association (GWAS) study to localize five IgAN susceptibility, in 10,775 individuals from Europe, Asia and America [93]. Nevertheless, these studies did not include patients with IgAV. There is only one small genetic study (285 IgAV patients and 1006 healthy controls from Spain genotyped by Illumina HumanCore BeadChips) showing in IgAV, as for IgAN, variations in the loci of the HLA class 2 genes region (HLA-DRB1 position 13 and 11) suggesting the same susceptibilities [94].

## 8. Discussion and Future Research

After reporting all those studies, can we say that IgA Nephropathy and IgA Vasculitis are two clinical entities of the same disease?

Clinical studies showed that they differentiate clearly in terms of clinical presentation and age at onset.

Concerning outcome, studies are conflicting, but tend to show that if patients are stratified on age and genetic background, IgAN and IgAV have the same renal prognosis. The presence of clinically speaking extra-renal disease makes the diagnosis of IgAV easy at an early stage, whereas in patients whose disease is limited to the kidney, the diagnosis is inevitably belated and therefore more advanced. It is not said, moreover, that these patients had, some years before, some unnoticed purpuric lesions. Therefore, the real question is: Why do some patients with IgAV have no renal involvement, and why do patients with IgAN have no skin lesions?

Physiopathological mechanisms and their related biomarkers are similar, each time they have been evaluated in the same study, none of which have been identified, to date, as having a strong prognostic value, either in IgAN and IgAV. It is thus most essential to identify early diagnostic and prognostic markers, which could be able to detect patients who will not spontaneously heal and require specific treatment (yet to be defined). Working together to set up new clinical studies appears necessary. It will be crucial for those future trials:-To include both diseases;-To agree on a common histological classification. Thus far, in fact, there is no consensual renal histologic classification for IgAV. Although the International Study of Kidney Disease in Children (ISKDC) classification is widely use in child IgAV, it is more and more questioned because it does not completely correlate with the clinical presentation and long term renal outcome. Few teams have applied to IgAV the Oxford classification widely used now for IgAN and have shown discordant results [95,96,97,98]. Its prognostic interest is actually disputed. A large international study, based on the model, which has resulted in the Oxford classification, is currently being developed for the IgAV;-To stratify the cohorts on age and genetic background, which are, to date, the only prognostic factors so far clearly identified.

## 9. Conclusions

Since the last reviews, published more than 10 years ago now, several clinical studies, reported here, provide additional arguments that IgAN and IgAV would be the same disease. In the absence of large studies, including adults and children from different geographical part of the world, suffering from IgAN or IgAV with or without renal impairment, it is not yet possible to conclude on their differences or similarity in terms of prognosis and sensitivity to treatment. Answering these questions gives opportunity to future clinical studies.

## Figures and Tables

**Figure 1 jcm-10-02310-f001:**
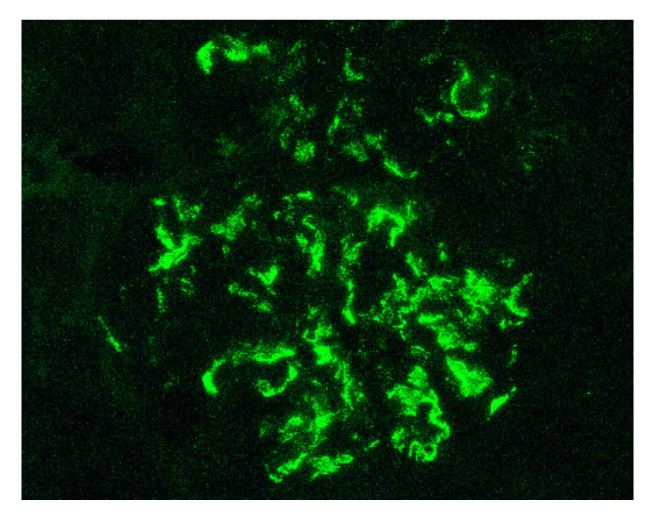
Mesangial and capillary wall IgA deposits (immunofluorescence staining for IgA, original magnification ×400).

**Figure 2 jcm-10-02310-f002:**
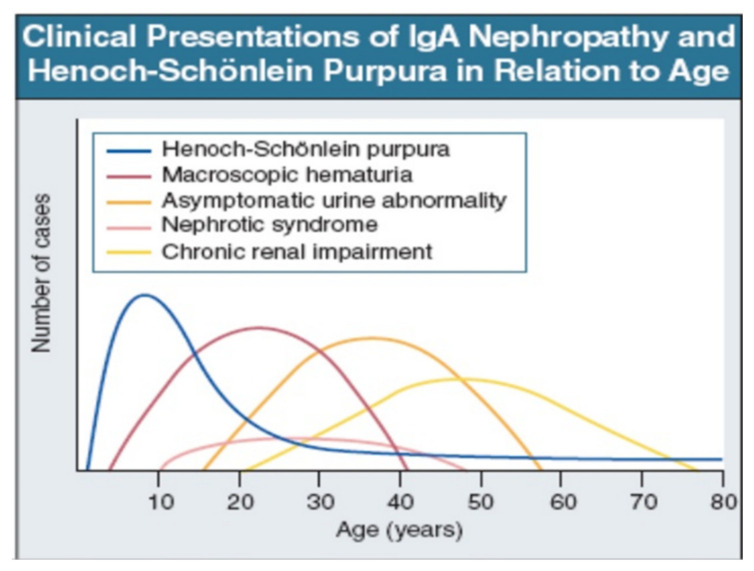
IgAN and IgAV clinical presentation over time from childhood to older age [35]. Reprinted with permission from ref. [35]. Copyright 2000 Rights and Permissions of Elsevier.

**Figure 3 jcm-10-02310-f003:**
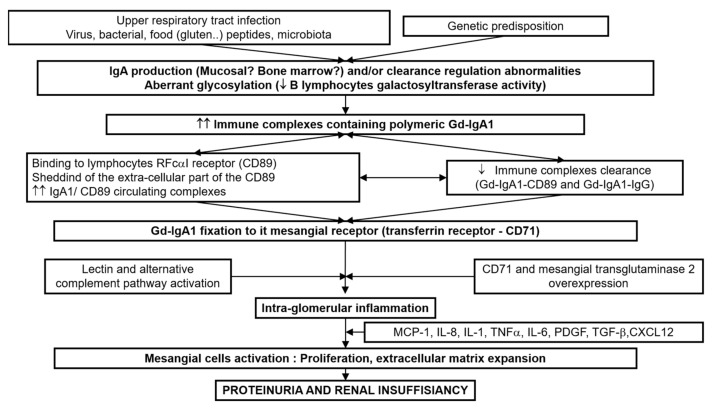
Physiopathology mechanism.

**Table 1 jcm-10-02310-t001:** Differences and similarities between IgAN and IgAV.

	IgA Nephropathy	IgA Vasculitis
Age at onset	30 to 39 years	1 to 19 years and 60 to 69 years
Clinical presentation	Only renal	Extra-renal symptoms (skin, gastro-intestinal, joint, neurologic, pulmonary, urologic) ± renal involvement
Renal biopsy	Mesangial IgA1, IgG, IgM, C3 and fibrin on immunofluorescence Mesangial hyper-cellularity with increased mesangial matrix, endo-capillary hyper-cellularity, segmental glomerular sclerosis, cellular crescents on light microscopy	
Outcome	More severe in adults	
Physiopathology	Multi-hit model involving IgA1	

## Data Availability

Not applicable.

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
