# Peer review of "IgA Vasculitis and IgA Nephropathy: Same Disease?"

_jcm, 2021, doi:10.3390/jcm10112310_

Round 1

Reviewer 1 Report

I congratulate the author for this detailed manuscript. The review is quite comprehensive, however does not answer the question in the conclusion part. I would like the author to address the following:

  1. Line 28: starts with 'Man', not sure what the author was trying to say here. Please reframe the opening sentence. The second sentence starts with 'they', its unclear who these 'they' are, please be specific.
  2. Line 30: need to remove citation part written in parathesis.
  3. line 32: would suggest to remove the word 'etc'
  4. Line 56: change depends of to depends 'on'
  5. Line 56: did you mean microscopic hematuria detection?
  6. Line 59: remove the parenthesis part.
  7. Line 63: change 'japan study' to Japanese study
  8. to maintain uniformity, please use either IgA nephropathy or IgA nephritis. Also, please use abbreviations as IgAN or IgAV respectively, to maintain consistency.
  9. I would recommend to describe the results of the studies in 'past tense', and not present tense as many results are different for differenet studies, and stating them in present tense is more suggestive of a fact, which clearly can not be true in state of conflicting results
  10. line 80: Remove EULAR
  11. line 82-85: starting from presence of......it is unclear what you are trying to state here. Kindly specify if you are describing the histology on light microscopy. Would also suggest to add an image of the histological findings to get more attention from the readers.
  12. Line 97: replace nephritic syndrome with nephritis
  13. Line 98: replace nephritic syndrome with glomerulonephritis
  14. Line 113: please reframe the sentence....'this last furthermore' does not sound right.
  15. Line 122: say 'kidney biopsy proven' instead of biopsy proven unless it was a different tissue being biopsied
  16. line 122: replace igA with IgAV
  17. line 174: replace prognosis with prognostic
  18. line 177: replace term with terms
  19. Lines 192-218: I highly recommend to put this in a table form or consider deleting this altogether, as I am unsure of its importance to the readers
  20. Conclusion: the research question that has been proposed in the abstract remains unanswered. Please expand the conclusion part, and/ or consider adding a discussion section prior to conclusion, to describe in more details, what your inference is, based on the literature you have described. While the studies have been described in great detail by you, it remains unclear what you are trying to suggest. 
  21. Please add a paragraph regarding research opportunities/ future prospects in this field.
  22. Adding a table figure/ table describing the similarities and differences beteween the 2 disease entities will be very helpful, and also throw more light on the question of these being 2 different entities or just 2 different manifestations of the same underlying disease process.
  23. Is there any data available to suggest preventive measures for development of nephritis, in patients with IgAV? If so, please include that in the manusctipt as well.
  24. You have given your opinion on certain studies in different sections, I would recommend to put that all together in the discussion section.
  25. Line 379: change word during to 'for'??
  26. Line 380: please do not say 'I believe', please be straightforward to state whether there is an international study going on, if not, jsut consider saying an international study is warranted. 

Author Response

Dear Editors,

We are submitting a revised version of our manuscript entitled IgA Vasculitis and IgA Nephropathy: Same disease?” previously entitled “IgA Vasculitis and IgA Nephritis: Same disease?" for publication in Journal of Clinical Medicine.

We thank the reviewers for their cogent remarks. We have provided point-by-point responses to the reviewers and made the corresponding modifications in the manuscript. We hope that these modifications will satisfy the Editorial Board of the journal and that the paper will be found to be suitable for publication. We stand ready to respond to any queries that may arise.

Sincerely yours.

Evangéline Pillebout

Reviewer n°1

I changed and corrected each of the following points, as suggested

- Line 28: starts with 'Man', not sure what the author was trying to say here. Please reframe the opening sentence. The second sentence starts with 'they', its unclear who these 'they' are, please be specific.

- Line 30: need to remove citation part written in parathesis.

- line 32: would suggest to remove the word 'etc'

- Line 56: change depends of to depends 'on'

- Line 56: did you mean microscopic hematuria detection?

- Line 59: remove the parenthesis part.

- Line 63: change 'japan study' to Japanese study

- to maintain uniformity, please use either IgA nephropathy or IgA nephritis. Also, please use abbreviations as IgAN or IgAV respectively, to maintain consistency.

- I would recommend to describe the results of the studies in 'past tense', and not present tense as many results are different for differenet studies, and stating them in present tense is more suggestive of a fact, which clearly can not be true in state of conflicting results

- line 80: Remove EULAR

- Line 97: replace nephritic syndrome with nephritis

- Line 98: replace nephritic syndrome with glomerulonephritis

- Line 113: please reframe the sentence....'this last furthermore' does not sound right.

- Line 122: say 'kidney biopsy proven' instead of biopsy proven unless it was a different tissue being biopsied

- line 122: replace igA with IgAV

- line 174: replace prognosis with prognostic

- line 177: replace term with terms

- Line 379: change word during to 'for'??

- Line 380: please do not say 'I believe', please be straightforward to state whether there is an international study going on, if not, jsut consider saying an international study is warranted. 

- line 82-85: starting from presence of......it is unclear what you are trying to state here. Kindly specify if you are describing the histology on light microscopy. Would also suggest to add an image of the histological findings to get more attention from the readers. I modify the sentence (“Renal biopsy shows, in the two cases: on immunofluorescence, predominant IgA1 deposits in the mesangium of all glomeruli (Figure 1), with glomerular deposits of IgG, IgM, C3 and fibrin in variable proportions; on light microscopy, mesangial hypercellularity with increased mesangial matrix, endo-capillary hypercellularity, segmental glomerular scle-rosis, cellular crescents and tubular atrophy and interstitial fibrosis“) and add a figure showing an IgA mesangial staining on immunofluorescence (Figure 1)

Figure 1. Mesangial and capillary wall IgA deposits (immunofluorescence staining for IgA, original magnification x400).

- Lines 192-218: I highly recommend to put this in a table form or consider deleting this altogether, as I am unsure of its importance to the readers. I deleted the paragraph

- Adding a table figure/ table describing the similarities and differences beteween the 2 disease entities will be very helpful, and also throw more light on the question of these being 2 different entities or just 2 different manifestations of the same underlying disease process. I add a table (Table 1) showing the main differences and similarities according to what can be retained from literature. I didn’t include in the table “treatment and genetic”, as far as we don’t have any clinical study comparing IgA and IgAV.

IgA Nephropathy

IgA Vasculitis

Age at onset

30 to 39 years

1 to 19 years and 60 to 69 years

Clinical presentation

Only renal

Extra-renal symptoms (skin, gastro-intestinal, joint, neurologic, pulmonary, urologic) +/- renal involvement

Renal biopsy

Mesangial IgA1, IgG, IgM, C3 and fibrin on immunofluorescence

Mesangial hyper-cellularity with increased mesangial matrix, endo-capillary hyper-cellularity, segmental glomerular sclerosis, cellular crescents on light microscopy

Outcome

Wild variation - More severe in adults

Physiopathology

Multi-hit model involving IgA1

- Is there any data available to suggest preventive measures for development of nephritis, in patients with IgAV? If so, please include that in the manusctipt as well. I add a paragraph to answer this question : “Two randomized placebo-controlled prednisone trials [65] [62] and one meta-analysis [61] showed that corticosteroids are ineffective to prevent occurrence of nephritis in children with IgAV. One ongoing study (NCT04008316) will evaluate colchicine in adult patients with IgAV limited to skin to prevent skin relapse (primary endpoint) and occurrence of digestive or kidney involvement (secondary endpoint)”

- Conclusion: the research question that has been proposed in the abstract remains unanswered. Please expand the conclusion part, and/ or consider adding a discussion section prior to conclusion, to describe in more details, what your inference is, based on the literature you have described. While the studies have been described in great detail by you, it remains unclear what you are trying to suggest. 

- Please add a paragraph regarding research opportunities/ future prospects in this field.

- You have given your opinion on certain studies in different sections, I would recommend to put that all together in the discussion section.

I rewrite and split the conclusion to detail more my analysis of the studies and address the topic of future studies

Discussion and future research

After reporting all those studies, can we say that IgA Nephropathy and IgA Vasculitis are two clinical entities of the same disease?

Clinical studies showed that they differentiate clearly in terms of clinical presentation and age at onset.

Concerning outcome, studies are conflicting but tend to show that if patients are stratified on age and genetic background IgAN and IgAV have the same renal prognosis. The presence of clinically-speaking extra-renal disease makes the diagnosis of IgAV easy at an early stage, whereas in patients whose disease is limited to the kidney, the diagnosis is inevitably belated and therefore more advanced. It is not said, moreover, that these patients had, some years before, some unnoticed purpuric lesions. So, the real question is: Why do some patients with IgAV have no renal involvement and why do patients with IgAN have no skin lesions?

Physiopathological mechanisms and their related biomarkers are similar, each time they have been evaluated in the same study, none of which have been identified, to date, having a strong prognostic value, either in IgAN and IgAV. It is thus most essential to identify early diagnostic and prognostic markers, which could be able to detect patients who will not spontaneously heal and require specific treatment (yet to be de-fined). Working together to set up new clinical studies appears necessary. It will be crucial for those future trials:

  • To include both diseases;
  • To agree on a common histological classification. So far, in fact, there is not any consensual renal histologic classification for IgAV. Although the International Study of Kidney Disease in Children (ISKDC) classification is widely use in children IgAV, it is more and more questioned because it doesn’t completely correlate with the clinical presentation and long term renal outcome. Few teams have applied to IgAV the Oxford classification widely used now for IgAN and have shown discordant results [95–97]. Its prognostic interest is actually disputed. A large international study, based on the model which has resulted in the Oxford classification, is currently being developed for the IgAV;
  • To stratify the cohorts on age and genetic back-ground, which are, to date, the only prognostic factors so far clearly identified.

Conclusion

Since the last reviews, published more than 10 years ago now, several clinical studies, reported here, provide additional arguments that IgA N and IgAV would be the same disease. In the absence of large studies, including adults and children from different geographical part of the world, suffering from IgAN or IgAV with or without renal impairment, it is not yet possible to conclude on their differences or similarity in terms of prognosis and sensitivity to treatment. Answering these questions gives opportunity to future clinical studies”

To detail more my analysis and address the topic of future studies

Pour détailler davantage mon analyse et aborder le sujet des futures études

for more details of my analysis and approach the subject of future studies

Pour plus de détails sur mon analyse et abordez le sujet des futures études

Impossible de charger les résultats complets

Réessayer

Nouvel essai…

Nouvel essai…

Reviewer 2 Report

The author describes recent concept and relation of IgA vasculitis and IgA nephropathy. The manuscript of interesting to readers. A couple of concerns can be more discussed. Comments are as follows: 

  1. Title : IgA nephritis→ IgA nephropathy ?
  2. The main question, IgA vasculitis and IgA nephropathy are the same disease?, is very interesting for readers including me. However, in clinical aspects, it seems that it may be still difficult to determine these 2 diseases are the same disease/origin. As describes in this paper, the timing of disease onset is quite different and overlapping these 2 diseases are quite rare, that may imply the mechanisms of 2 diseases is different, even though renal histology is indistinguishable. The most thing I wonder, co-occurrence of IgA nephropathy and lesions of vasculitis seen in IgA vasculitis (HSP) is quite rare, or hardly observed (I never experienced so far). Indication of immunosuppressive therapy between IgA vasculitis (aggressive) and IgA nephropathy (?) is also of concern that the mechanism of disease onset is quite different. I'm very glad if we can get clear answer. 
  3. Physiopathology: It would be very helpful if the author gives schematic illustration of recent concept regarding pathogenesis of IgA nephropathy/IgA vasculitis. 
  4. Biomarker: What imply the X in the Table 1? Please re-make more understandable.    

Author Response

Dear Editors,

We are submitting a revised version of our manuscript entitled IgA Vasculitis and IgA Nephropathy: Same disease?” previously entitled “IgA Vasculitis and IgA Nephritis: Same disease?" for publication in Journal of Clinical Medicine.

We thank the editors for their cogent remarks. We have provided point-by-point responses to the reviewers and made the corresponding modifications in the manuscript. We hope that these modifications will satisfy the Editorial Board of the journal and that the paper will be found to be suitable for publication. We stand ready to respond to any queries that may arise.

Sincerely yours.

Evangéline Pillebout

Reviewer n°2

  • Title : IgA nephritis→ IgA nephropathy ? I change the title: “IgA Vasculitis and IgA Nephropathy: Same disease?”
  • The main question, IgA vasculitis and IgA nephropathy are the same disease?, is very interesting for readers including me. However, in clinical aspects, it seems that it may be still difficult to determine these 2 diseases are the same disease/origin. As describes in this paper, the timing of disease onset is quite different and overlapping these 2 diseases are quite rare, that may imply the mechanisms of 2 diseases is different, even though renal histology is indistinguishable. The most thing I wonder, co-occurrence of IgA nephropathy and lesions of vasculitis seen in IgA vasculitis (HSP) is quite rare, or hardly observed (I never experienced so far). Indication of immunosuppressive therapy between IgA vasculitis (aggressive) and IgA nephropathy (?) is also of concern that the mechanism of disease onset is quite different. I'm very glad if we can get clear answer

  • Like you, I wish I could settle definitively, and clearly decide how to manage these patients. This review gives some answers. I rewrite the discussion/conclusion part to be more intelligible.

Discussion and future research

After reporting all those studies, can we say that IgA Nephropathy and IgA Vasculitis are two clinical entities of the same disease?

Clinical studies showed that they differentiate clearly in terms of clinical presentation and age at onset.

Concerning outcome, studies are conflicting but tend to show that if patients are stratified on age and genetic background IgAN and IgAV have the same renal prognosis. The presence of clinically-speaking extra-renal disease makes the diagnosis of IgAV easy at an early stage, whereas in patients whose disease is limited to the kidney, the diagnosis is inevitably belated and therefore more advanced. It is not said, moreover, that these patients had, some years before, some unnoticed purpuric lesions. So, the real question is: Why do some patients with IgAV have no renal involvement and why do patients with IgAN have no skin lesions?

Physiopathological mechanisms and their related biomarkers are similar, each time they have been evaluated in the same study, none of which have been identified, to date, having a strong prognostic value, either in IgAN and IgAV. It is thus most essential to identify early diagnostic and prognostic markers, which could be able to detect patients who will not spontaneously heal and require specific treatment (yet to be de-fined). Working together to set up new clinical studies appears necessary. It will be crucial for those future trials:

  • To include both diseases;
  • To agree on a common histological classification. So far, in fact, there is not any consensual renal histologic classification for IgAV. Although the International Study of Kidney Disease in Children (ISKDC) classification is widely use in children IgAV, it is more and more questioned because it doesn’t completely correlate with the clinical presentation and long-term renal outcome. Few teams have applied to IgAV the Oxford classification widely used now for IgAN and have shown discordant results [95–97]. Its prognostic interest is actually disputed. A large international study, based on the model which has resulted in the Oxford classification, is currently being developed for the IgAV;
  • To stratify the cohorts on age and genetic back-ground, which are, to date, the only prognostic factors so far clearly identified.

  1. Like you, I would like to be able to definitely decide and clearly decide how to support these patients.
  2. Comme vous, j'aimerais pouvoir décider définitivement et décider clairement comment soutenir ces patients.
  3. Like you, I wish I could settle definitively and clearly decide how to manage these patients.
  4. Comme vous, j'aimerais pouvoir régler définitivement et clairement décider comment gérer ces patients.
  5. Impossible de charger les résultats complets
  6. Réessayer
  7. Nouvel essai…
  8. Nouvel essai…
    1. Physiopathology: It would be very helpful if the author gives schematic illustration of recent concept regarding pathogenesis of IgA nephropathy/IgA vasculitis. I add a figure, Figure 3: Multi-hit pathogenesis model for glomerulonephritis in IgAN and IgAV
    2.  
  • Biomarker: What imply the X in the Table 1? Please re-make more understandable. I change the table (now Table 2), including in each column, the references in which are studied each biomarkers. I hope this is more understandable

IgAN

IgAV

IgAN + IgAV

GdIgA1

[75] [76]

[74] [77] [78]

[73] [40]

GdIgA1/sCD89

[79]

[77][78]

GdIgA1/IgG

[80]

[77][78]

sCD89

[81]

Transglutaminase2

[82]

CD71

[82]

TLR9

[83,84]

[67]

TLR4

[66,67]

TGF-β1 MCP1

[40]

Complement system

[69][72]

[85]

Reviewer 3 Report

In the present review, the author try to evaluate, according to current knowledge, wether IgA vasculitis and IgA nephritis are the two sides of the same pathology.

To this end, E. Pillebout draw the story of these pathologies (or this pathology?) through the classical way of "Epidemiological, clinical, biomarkers and treatment"...

This approach has already been proposed in other reviews, that clearly evaluates IgA vasculitis and nephropathy similarities and differences.

So, in my opinion, the author should state instead of another similar approach of these diseases: "IgA Vasculitis and IgA Nephritis: new elements to consider them as a unique disease?".

Then, the paper must be shortened. The authors should focus on new elements that help us to understand the disease.

  • section clinical presentation and outcomes: instead of this too long section, a Table could summarize main results of the presented studies, only when they add some new relevant information.
  • Figure 1: the figure is not mentioned is the text, please correct this. The resolution of the figure is poor. This figure appears to be like a print-screen. In the acknowledge section, only the author of the figure is identify. Where does this figure come from? please provide a figure of decent quality...
  • "Treatment section": I am note sure that the copy-paste of KDIGO guidelines is useful... This paragraph cost more than 30 lines for the reader... Is it relevant in the field of this review? 
  • In my opinion, the epidemiological, clinical and treatment sections must be largely shortened...
  • Physiopathology section and biomarkers: isn't here the points that must be stressed out? When large epidemiological, clinical and datas issues from human trial don't give answer, physicians and researchers found in fundamental research a way to better understand pathophysiology of the disease. 
    Is there relevant animal model of the disease(s)? Was a new approach developed? The better understanding of the pathophysiology should be presented, and it must be detailed wether mechanism(s) are described in IgA Nephropathy or IgA Vasculitis? 
  • The same for biomarkers section.

If new approach of the pathophysiology of the disease are in accordance with the hypothesis that IgA nephropathy and vasculitis are the same disease, isn't it what must be emphasized in such paper?

Finally, as far as I know, few is new in the understanding of the disease, is a review needed, or an article in a letter format may be enough? 

Author Response

Dear Editors,

We are submitting a revised version of our manuscript entitled IgA Vasculitis and IgA Nephropathy: Same disease?” previously entitled “IgA Vasculitis and IgA Nephritis: Same disease?" for publication in Journal of Clinical Medicine.

We thank the editors for their cogent remarks. We have provided point-by-point responses to the reviewers and made the corresponding modifications in the manuscript. We hope that these modifications will satisfy the Editorial Board of the journal and that the paper will be found to be suitable for publication. We stand ready to respond to any queries that may arise.

Sincerely yours.

Evangéline Pillebout

Reviewer n°3

In the present review, the author try to evaluate, according to current knowledge, wether IgA vasculitis and IgA nephritis are the two sides of the same pathology.

To this end, E. Pillebout draw the story of these pathologies (or this pathology?) through the classical way of "Epidemiological, clinical, biomarkers and treatment"...

This approach has already been proposed in other reviews, that clearly evaluates IgA vasculitis and nephropathy similarities and differences.

So, in my opinion, the author should state instead of another similar approach of these diseases: "IgA Vasculitis and IgA Nephritis: new elements to consider them as a unique disease?".

Then, the paper must be shortened. The authors should focus on new elements that help us to understand the disease.

    • section clinical presentation and outcomes: instead of this too long section, a Table could summarize main results of the presented studies, only when they add some new relevant information. Please refer to the last answer.
    • Figure 1: the figure is not mentioned is the text, please correct this. The resolution of the figure is poor. This figure appears to be like a print-screen. In the acknowledge section, only the author of the figure is identify. Where does this figure come from? please provide a figure of decent quality... I provide a high quality figure
  • "Treatment section": I am note sure that the copy-paste of KDIGO guidelines is useful... This paragraph cost more than 30 lines for the reader... Is it relevant in the field of this review? I deleted the paragraph
  • In my opinion, the epidemiological, clinical and treatment sections must be largely shortened...Please refer to the last answer.
  • Physiopathology section and biomarkers: isn't here the points that must be stressed out? When large epidemiological, clinical and datas issues from human trial don't give answer, physicians and researchers found in fundamental research a way to better understand pathophysiology of the disease. 
    Is there relevant animal model of the disease(s)? Was a new approach developed? The better understanding of the pathophysiology should be presented, and it must be detailed wether mechanism(s) are described in IgA Nephropathy or IgA Vasculitis? There is to date no relevant animal model in IgAV; I added a figure to better detail the pathophysiology (Figure 3)
  • The same for biomarkers section.

If new approach of the pathophysiology of the disease are in accordance with the hypothesis that IgA nephropathy and vasculitis are the same disease, isn't it what must be emphasized in such paper?

Finally, as far as I know, few is new in the understanding of the disease, is a review needed, or an article in a letter format may be enough? 

I fully understand your remarks and my article could indeed hold in one page.

Nevertheless, the invitation of the publisher was a review of at least 5 thousand words.

Therefore, I wrote a detailed, didactic article, with the aim of reporting published studies on the subject since the last review published over 10 years ago

Certainly, I do not answer the question “IgAV and IgAN same disease?”, but I bring all the information that the reader would need so that he can make his own opinion. I rewrote the discussion/conclusion to be more incisive, but as far as so few clinical trials evaluate the both together to compare outcome, sensibility to treatment… I cannot really give a definitive conclusion. I propose tracks for future studies or research opportunities that would do it.

Round 2

Reviewer 3 Report

Thank you for your work to provide this revised version of the manuscript.

"Nevertheless, the invitation of the publisher was a review of at least 5 thousand words.

Therefore, I wrote a detailed, didactic article, with the aim of reporting published studies on the subject since the last review published over 10 years ago"

As a reviewer, I was unaware of the "invitation" nature of this work. However, my opinion remain the same.

I find the idea of the new Figure for the possible picture of the involved pathophysiological mechanisms interesting. However, the quality of the picture is low, and I can't read it easily. 

Elsewhere, I think that the quality of the figure 2 remains insufficient to be published in the journal... 

I have no other comment.

Author Response

Thank you very much for your comments. I have provided to the editor a high-quality version of each of the 3 figures. Obviously they do not have sent you. I apologize for this. I assure you that the reader will have a good version of the images.